# SOLA: Text-based animated vector graphics generation with agentic orchestration

## Abstract

We introduce SOLA (SVG-Orientied Language-to-Animation), a novel end-to-end generative pipeline that produces animated scalable vector graphics (SVGs) directly from natural language prompts. Unlike prior systems such as Keyframer, which requires a static SVG input image and uses GPT-4 to generate CSS animations for that image, our approach constructs the entire animated SVG sequence from scratch. SOLA employs an agentic pipeline architecture (LangGraph) to orchestrate multiple modules and ensure coherent results. Given a text description, it first synthesizes an initial sequence of video frames, then vectorizes each frame into SVG path shapes, and aligns corresponding shapes across frames via greedy bipartite matching. We normalize all shape outlines to a consistent polyline representation and convert them into smooth cubic Bezier curves for smooth morphing between frames. This shape-level processing is the key to a resolution-independent animation with coherent motion. To overcome the absence of existing benchmarks for text-to-SVG animation, we design a thorough evaluation protocol with a prompt test set and diverse performance metrics. Experimental results with this protocol demonstrate the superiority of our approach compared to state-of-the-art LLM-based methods in translating high-level language descriptions into fully vectorized animations.

## 1 Introduction

Text-driven generation of visual content has seen remarkable progress, from high-fidelity images to short videos (Esser et al., 2023). However, extending this success to vector graphics and animations poses unique challenges. Vector graphics (e.g. SVGs) are valued for their resolution-independence and compactness, making them ideal for icons, web graphics, and scalable art (Jain et al., 2023). However, unlike pixel images, large datasets of SVG graphics (especially animated SVGs) are scarce and often encumbered by copyright restrictions, making it difficult to train or fine-tune generative models directly on vector data.

Prior approaches to vector graphic synthesis have largely fallen short of tackling text-to-SVG *animation*. DeepSVG (Carlier et al., 2020) introduces a hierarchical generative model for SVG icons and demonstrates animation only via latent-space interpolation, rather than text-driven animation. VectorFusion (Jain et al., 2023) abstracts pixel-based diffusion models with a differentiable rasterizer to produce *static* SVGs, not animations. NeuralSVG (Polaczek et al., 2025) focuses on text-to-vector generation with layered SVG structure, and AnimateSVG (Mateja et al., 2023) automates logo animations within a domain-specific pipeline guided by aesthetics modeling; neither produces animated SVGs directly from natural language. Thus, despite the progress in vector generation, there has not yet been a clear breakthrough in generating SVG animations directly from text.

In this work, we introduce **SOLA** (*SVG-Oriented Language-to-Animation*), the first pipeline to generate animated SVGs directly from natural language prompts without any additional training or fine-tuning. In contrast to the prior art, our approach leverages only pre-trained models and an agentic workflow to bridge text prompts to vector animations. Specifically, we orchestrate a sequence of off-the-shelf generative models and vector processing steps using a LangGraph agent. Given a user prompt, the system first produces an initial raster animation using state-of-the-art text-to-image and text-to-video generation (e.g., Runway's Gen-4 model, which can synthesize short videos from

a text description [1]). This video is then automatically converted into an SVG animation through a series of shape-level operations – all performed with no task-specific training. By decomposing the problem and tapping into powerful pre-trained components, our pipeline achieves text-to-vector animation on arbitrary prompts in a zero-shot manner.

Our main contributions can be summarized as follows:

- **Zero-shot text-to-SVG animation pipeline:** We present the first agentic workflow using LangGraph [2] that orchestrates pre-trained models (GPT-5 [3], GPT-Image-1 [4], Gen-4 [5], VTracer [6]) to generate animated SVGs from text without training.

- **Shape tracking via bipartite matching:** We introduce feature-based shape correspondence (Algorithm 1) using greedy matching on centroid, bbox, and perimeter features. This maintains consistent object identity across frames, achieving significantly improved motion smoothness compared to baselines.

- **Smooth morphing with path normalization:** Our arc-length resampling and RMS-based circular alignment (Algorithm 2) enables SMIL-based shape interpolation with significantly lower jerk, producing fluid animations without learned models.

- **Comprehensive evaluation framework:** Due to the absence of existing benchmarks for text-to-SVG animation, we establish an 80-prompt test set spanning three complexity levels and introduce systematic evaluation metrics including temporal smoothness, animation complexity, and semantic alignment via structured LLM-as-Judge (Zheng et al., 2023; Chen et al., 2024) with a 100-point rubric across four assessment dimensions.

This design demonstrates a practical and scalable solution for high-quality SVG animation generation from language alone. By harnessing existing generative models and carefully orchestrating vector operations, we sidestep the need for large curated SVG animation datasets or model retraining. The resulting system can produce resolution-independent, animations for a wide range of prompts – something previously unattainable with purely text-driven methods.

## 2 RELATED WORK

**Data-driven SVG generation and animation.** Early approaches to learning-based vector graphics generation, such as Sketch-RNN (Ha & Eck, 2017), learned to produce simple vector drawings from large sketch datasets but were constrained to crude strokes and fixed object categories. Subsequent methods including SVG-VAE (Lopes et al., 2019) and DeepSVG (Carlier et al., 2020) improved generation fidelity by leveraging variational autoencoders and Transformers to synthesize more complex icons and fonts. However, these approaches required curated vector training data and often limited outputs to basic path primitives, resulting in poor generalization beyond their training domains. To alleviate the dependency on vector-format supervision, hybrid raster-vector techniques emerged. Im2Vec (Reddy et al., 2021), for instance, demonstrated vector graphics synthesis by differentiably rendering predicted shapes to match input images, enabling training on ordinary pixel art without SVG ground truth. Meanwhile, text-guided approaches utilizing vision-language models (e.g., CLIPDraw (Frans et al., 2022)) and diffusion pipelines (e.g., DiffSketcher (Xing et al., 2023)) have been applied to SVG generation, yet their outputs often consist of tangled, uneditable paths and remain limited to static single-frame graphics. Structured two-stage pipelines like Chat2SVG (Wu et al., 2025) combine LLM planning with diffusion-based refinement to enhance output detail, but even these produce only single-frame illustrations. The challenge of text-driven SVG animation—generating multi-frame vector graphics from prompts—remains largely unaddressed. While tools like Keyframer (Tseng et al., 2024) can animate existing SVGs via GPT-4-generated CSS instructions and AnimateSVG (Mateja et al., 2023) automatically generates animated logos, these approaches either require pre-existing static artwork or are inherently domain-specific, failing to generalize to novel scenes or produce complex multi-object SVG animations from text alone.

---

[1] https://runwayml.com/research/introducing-runway-gen-4
[2] https://github.com/langchain-ai/langgraph
[3] https://openai.com/index/introducing-gpt-5/
[4] https://openai.com/index/introducing-4o-image-generation/
[5] https://runwayml.com/research/introducing-runway-gen-4
[6] https://www.visioncortex.org/vtracer/

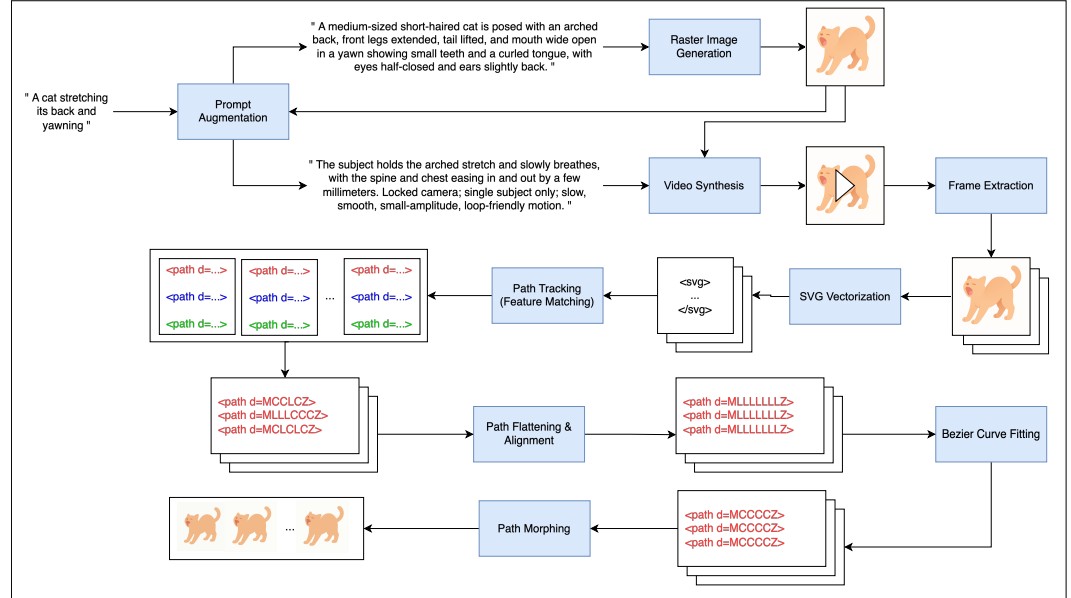

Figure 1: Overall architecture of our text-to-SVG animation pipeline. The system transforms a text prompt into an animated SVG through sequential stages: prompt augmentation, raster image generation, video synthesis, frame extraction, SVG vectorization, and path processing (tracking, alignment, morphing). The final Bezier curve fitting step produces smooth, resolution-independent animations.

**Agentic AI frameworks for complex task orchestration.** Recent advances in agentic AI have enabled language models to orchestrate complex, multi-step workflows more effectively, with recent work extending agentic workflows to domain-specific challenges in chemistry (Pham et al., 2025), robotics (Moncada-Ramirez et al., 2025), and medical imaging (Kim et al., 2025). Techniques such as function calling [7] allow LLMs to invoke external tools or APIs as needed, while higher-level frameworks including AutoGen (Wu et al., 2024) and LangGraph provide structured paradigms for coordinating multiple specialized sub-agents in decomposed tasks. These approaches enhance robustness by dividing large problems into modular subtasks handled by expert modules, demonstrating success in domains ranging from code generation to long-horizon planning. However, general orchestration frameworks alone do not solve domain-specific creative problems; they provide infrastructure, but effective strategies and domain knowledge must be designed by the system creator.

## 3 METHOD

### 3.1 PIPELINE OVERVIEW AND AGENTIC ORCHESTRATION

We implement our text-to-animation pipeline as an agentic LangGraph workflow, where each step is a node controlled by an LLM-based agent. The LangGraph framework treats AI models and tools as stateful agents in a directed graph, enabling multi-step reasoning with shared context. This design allows the system to autonomously coordinate image generation, video synthesis, and vector processing in sequence.

Figure 1 illustrates the complete architecture of our text-to-SVG animation pipeline, showing how the framework orchestrates the transformation from natural language input to animated vector graphics output. Each step outputs intermediate artifacts (image, video, SVG frames, etc.) stored in the pipeline state and consumed by the next node. This modular graph design ensures agent autonomy in coordinating complex tasks (similar to recent multi-tool AI workflows). In contrast to prior interac-

---

[7] https://openai.com/blog/function-calling-and-other-api-updates

tive tools like Keyframer (Tseng et al., 2024), which rely on a user guiding a single LLM to generate animation code from a given static SVG, our method fully automates the prompt-to-animated-SVG conversion via an orchestrated pipeline.

### 3.2 PROMPT AUGMENTATION FOR VISUAL GENERATION

Given a natural-language prompt $P$, the proposed pipeline employs a multi-stage refinement strategy that produces specialized prompts for image generation ($P_{\text{img}}$) and video generation ($P_{\text{vid}}$), each optimized for their respective modality constraints.

**Static Scene Description for Image Generation** For image generation, we leverage a pre-trained LLM to transform potentially ambiguous user inputs into precise visual descriptions. The refinement process extracts subject-only descriptions focusing on static physical attributes (shape, size, colors, pose) while explicitly avoiding motion-related language. The refined prompt is augmented with a comprehensive style guide that enforces vector-friendly constraints: stroke-free rendering, flat solid colors without gradients, round geometric shapes suitable for SVG representation, and single-subject isolation to ensure clean vectorization.

**Motion-Aware Video Prompt Synthesis** The video prompt generation implements a pattern-based motion extraction mechanism that identifies action cues from the original prompt across 13 predefined motion categories (bounce, wave, spin, jump, pulse, etc.) and environmental effects (dust, breeze, rain). The final video prompt composition follows established video generation guidelines with three key constraints: (1) slow, smooth motion with a small amplitude suitable for looping, (2) a locked camera perspective for spatial consistency, and (3) sanitization of numeric literals and special characters that could trigger unintended text rendering.

### 3.3 VIDEO GENERATION

We employ a state-of-the-art text-to-video generation model to generate short video clips from refined prompts. Our implementation uses Gen-4 model [8], producing temporally coherent frame sequences depicting the described scene. We typically generate 5 second clips at 24 FPS, sufficient to capture simple animations from single prompts.

The pipeline optionally leverages initial keyframe images for guided video synthesis. When the text-to-video model supports image conditioning, we use either user-supplied images or generate keyframes via text-to-image diffusion models (e.g., GPT-Image-1 [9]) as visual anchors constraining video style and content. After generation, we extract individual raster frames $F_t$ for vectorization.

### 3.4 SVG VECTORIZATION OF FRAMES

We convert each raster frame into vector graphics using the open-source VTracer tool [10], which approximates bitmaps with smooth colored shapes (SVG paths). For each frame $t$, the vectorizer performs edge detection and shape fitting to produce a collection of closed paths $\mathcal{S}^t = \{S_1^t, S_2^t, \ldots, S_{n_t}^t\}$ that reconstruct the frame's visuals. Each path has a fill color matching its corresponding image region.

After initial SVG generation, we flatten every vector shape into polylines by converting curves and complex path commands into straight-line segment sequences (using only SVG `M` (move) and `L` (line) commands). This yields a normalized representation where each shape $S_i^t$ is represented by an ordered vertex list along its contour, facilitating shape comparison and morphing across frames.

### 3.5 SHAPE ALIGNMENT AND PATH TRACKING

Once each frame is vectorized into SVG paths, we establish correspondences between shapes across consecutive frames to enable coherent morphing animations. Our approach formulates shape

---

[8]`https://runwayml.com/research/introducing-runway-gen-4`

[9]`https://openai.com/index/introducing-4o-image-generation/`

[10]`https://www.visioncortex.org/vtracer/`

matching as a bipartite assignment problem, building consistent tracks that maintain object identity throughout the animation sequence.

**Feature-based Shape Matching.** For each pair of shapes in frame $t$ and frame $t + 1$, we extract geometric features and compute a dissimilarity cost. Let $c_i = (c_{i,x}, c_{i,y})$ denote the centroid of shape $S_i^t$, and let $w_i$, $h_i$, and $P_i$ represent its bounding box width, height, and perimeter respectively. The feature-based cost between shapes $S_i^t$ and $S_j^{t+1}$ is defined as:

$$C_{ij} = \|c_i - c_j\| + \lambda_b(|w_i - w_j| + |h_i - h_j|) + \lambda_p|P_i - P_j| \tag{1}$$

where $\lambda_b = 0.25$ and $\lambda_p = 0.01$ are weighting parameters that prioritize centroid stability while incorporating size and perimeter differences as soft constraints.

**Greedy Bipartite Matching.** We employ a greedy matching strategy that assigns shapes between consecutive frames based on their feature distances. The matching process sorts all possible pairs $(i, j)$ by their cost $C_{ij}$ and greedily selects matches below a threshold (300.0 in our implementation, relative to the 1024x1024 canvas). This approach efficiently handles typical animations while maintaining computational efficiency. The detailed matching algorithm is presented in Algorithm 1 in Appendix A.

**Path Normalization and Alignment.** After establishing shape correspondences, we normalize the path representations to enable smooth morphing. Each SVG path is first converted to a dense polyline by sampling curves at regular intervals, then resampled to a uniform number of points using arc-length parameterization. For a path $\mathcal{P}$ with varying vertex count, we: (1) sample the path densely to capture its geometry: $V = \{v_1, v_2, ..., v_k\}$, (2) compute cumulative arc-lengths: $L_i = \sum_{j=1}^{i} \|v_j - v_{j-1}\|$, and (3) resample at uniform arc-length intervals to obtain $N$ points: $Q = \{q_1, ..., q_N\}$, where

$$N = \min\big(150, \max_{\mathcal{P} \in \mathcal{T}} |\text{vertices}(\mathcal{P})|\big)$$

and $\mathcal{T}$ denotes the set of all shape tracks in the animation. This ensures a consistent point count across tracks while preserving geometric details.

**RMS-based Circular Alignment.** To handle closed contours where the starting vertex may vary between frames, we employ circular alignment based on root-mean-square (RMS) distance minimization. For two point sequences $Q$ and $R$ with $N$ points each, we find the optimal shift $k^*$:

$$k^* = \arg\min_{k \in [0,N)} \text{RMS}(\text{Shift}(Q, k), R) \tag{2}$$

where the RMS distance between $A = (a_1, \ldots, a_N)$ and $B = (b_1, \ldots, b_N)$ is:

$$\text{RMS}(A, B) = \sqrt{\frac{1}{N} \sum_{i=1}^{N} \|a_i - b_i\|^2} \tag{3}$$

and $\text{Shift}(Q, k)$ circularly rotates the sequence by $k$ positions: $(q_k, q_{k+1}, ..., q_{N-1}, q_0, ..., q_{k-1})$. The complete alignment procedure is detailed in Algorithm 2.

**Track Visibility Management.** Our tracking system maintains a visibility matrix $\mathcal{V} \in \{0, 1\}^{|\mathcal{T}| \times n}$, where $|\mathcal{T}|$ denotes the number of tracks and $n$ represents the total number of frames in the animation sequence. This matrix records whether each track is visible in each frame. When a shape disappears (no match found), we mark it as invisible but retain its last known state for potential reappearance. This approach enables smooth fade-in/fade-out effects through SVG opacity animation. Formally, we define the opacity function as:

$$\text{opacity}(T_i, t) = \begin{cases} 1 & \text{if } \mathcal{V}[i, t] = 1 \\ 0 & \text{if } \mathcal{V}[i, t] = 0 \end{cases} \tag{4}$$

Here, $\text{opacity}(T_i, t)$ specifies the transparency of track $T_i$ at frame $t$. A value of $1$ means that the shape is fully visible (rendered with its assigned fill color), while a value of $0$ means that the shape

is completely transparent and does not appear in the frame. Intermediate values between $0$ and $1$ can be interpolated by the renderer to produce smooth fading transitions when a shape enters or leaves the scene.

## 3.6 Path Morphing and SVG Animation

With shape correspondences established and paths normalized, we construct the animated SVG. The core idea is to render shape morphing across frames by smoothly interpolating the path coordinates of each shape. For a given shape correspondence, let $v_k^t = (x_k^t, y_k^t)$ be the $k$-th vertex of the shape in frame $t$ and $u_k^{t+1} = (x_k^{t+1}, y_k^{t+1})$ the corresponding vertex in frame $t+1$. We define the interpolated vertex position as:

$$r_k(\tau) = (1 - \tau)v_k^t + \tau u_k^{t+1}, \quad 0 \leq \tau \leq 1. \tag{5}$$

All vertices undergo the same linear interpolation, producing a time-continuous morph of the entire shape. We embed this in the SVG using SMIL animations on the path's d attribute. The outcome is a single SVG file containing all path definitions and embedded animations that morph them over the duration of the clip.

## 3.7 Cubic Bézier Curve Fitting

As a post-processing step, we refine vector outlines by fitting smooth cubic Bézier curves. While flattened polylines are accurate, they often contain numerous short segments. We replace polylines with cubic Bézier segments approximating the same shapes, yielding compact, smooth path representations. This conversion reduces vertex and command counts while maintaining accuracy within chosen error tolerances.

## 4 Experiments

Due to the scarcity of existing benchmarks and comparable methods for text-to-SVG animation generation, we establish strong baselines using state-of-the-art LLMs in direct generation mode. We evaluate our multi-stage pipeline approach against these LLM-based single-prompt generation methods through comprehensive experiments on animation complexity, motion smoothness, and semantic alignment metrics. Our experimental design aims to perform proper comparison of the proposed workflow against what current best-in-class language models can achieve in a single pass, thereby validating the effectiveness of our structured pipeline approach for this novel task.

### 4.1 Experimental Setup

**Baseline Justification.** The text-to-SVG animation task lacks established benchmarks or prior work with publicly available implementations. To establish meaningful comparisons, we leverage the most capable contemporary LLMs (GPT-5 and Claude-4.1 Opus) as baselines, providing them with carefully crafted system prompts that include all necessary SVG animation specifications. This approach represents the current best practice for zero-shot SVG generation and serves as a strong upper bound for single-pass methods.

**Baselines.** SVG animations are directly generated from text prompts (see Appendix B for details):

- **GPT-5-Direct**: Single-pass SVG generation using GPT-5 with a comprehensive system prompt detailing animation requirements (1024×1024 canvas, path-only shapes, morphable d-attribute animation, loop constraints).
- **Claude-Direct**: Direct generation using Claude-4.1 Opus with the identical specifications.

**Test prompts.** We curate 80 text prompts spanning different motion complexities (see Appendix C for complete prompts):

- Simple motions (e.g., "A ball gently bouncing up and down"): 30 prompts

- Complex motions (e.g., "A cat stretching its back and yawning"): 30 prompts
- Interactive motions (e.g., "Two birds flying side by side, wings synchronizing"): 20 prompts

## 4.2 IMPLEMENTATION DETAILS

For our pipeline, we use GPT-5 for prompt refinement (temperature 0.4), GPT-Image-1 for image synthesis (1024×1024 resolution), RunwayML Gen-4 Turbo for video generation (960×960, 5 seconds), and VTracer for vectorization. Each generated clip has a duration of 5 seconds at 24 FPS (≈120 frames), which are subsequently vectorized and aligned into morphable SVG paths.

For fair comparison, GPT-5-Direct and Claude-Direct were both provided with identical system prompts specifying a 1024×1024 canvas, path-only SVG construction, and loopable `<animate>` attributes with matching `d` attribute structure. Both baseline models were evaluated in single-prompt mode without iterative refinement, representing their strongest zero-shot capability. The baselines were allowed up to 2000 tokens for generation with structured output format (JSON schema) to ensure valid SVG syntax.

## 4.3 EVALUATION METRICS

To evaluate the quality of generated SVG animations, we adopt a comprehensive evaluation framework capturing structural complexity, temporal smoothness, and semantic fidelity.

**Animation Complexity.** We measure the structural properties of generated SVGs through multiple dimensions:

- *Static Complexity* ($C_{static}$): Number of SVG paths, total path commands, and commands per path
- *Dynamic Complexity* ($C_{dynamic}$): Number of `<animate>` elements, total keyframes, and commands per frame
- *Complexity Score*: $C = \log_{10}(C_{static} + C_{dynamic})$

A high complexity score is considered as a result of sufficient and faithful visual details in the generated SVG animation, thus is preferred.

**Motion Smoothness.** We assess temporal coherence through kinematic analysis of shape centroids across frames:

- *Jerk* ($j$): Rate of acceleration change (third derivative)
- *Smoothness Score*: $S = \max(0, 100 - j/n)$

The smoothness score evaluates how smooth (or jerky) the motions are and whether the generated animation contains too few frames to produce smooth motions.

**Semantic Alignment.** We employ an *LLM-as-Judge* (Zheng et al., 2023) evaluation using GPT-5 with structured outputs to assess prompt-animation correspondence. The evaluation leverages both visual frames and technical SVG metadata for comprehensive assessment. The judge examines 5 evenly sampled frames (0%, 25%, 50%, 75%, 100%) rendered at 512×512 pixels, alongside SVG structural metadata including file size, path count, command complexity, and keyframe density.

The evaluation uses a 100-point rubric with four main criteria:

- *Technical Implementation* (40 points): Evaluates structural complexity and implementation quality using both visual evidence and SVG metadata. This includes frame density assessment via keyframe count (8 pts), path complexity from command distribution (6 pts), and element intricacy through file size and complexity scores (6 pts), plus technical precision (10 pts) and comprehensive execution (10 pts).

Table 1: Comparison of animation generation methods across different motion complexity levels. A higher metric value is better. Mean and standard deviation values are shown. Best results are marked in bold.

| Experiment | Method | Complexity | Smoothness | Semantic |
|---|---|---|---|---|
| **Simple** | GPT-5 | 1.71±0.21 | 81.8±28.8 | 61±10 |
| | Claude | 2.09±0.21 | 72.3±32.2 | 69±8 |
| | **Ours** | **4.54±0.26** | **93.5±4.5** | **71±10** |
| **Complex** | GPT-5 | 1.91±0.16 | 82.1±28.3 | 41±9 |
| | Claude | 2.32±0.14 | 78.3±29.5 | 48±9 |
| | **Ours** | **4.67±0.17** | **96.0±2.3** | **73±11** |
| **Interactive** | GPT-5 | 1.97±0.18 | 93.3±19.5 | 50±12 |
| | Claude | 2.30±0.16 | 79.4±27.0 | 68±7 |
| | **Ours** | **4.75±0.12** | **93.7±4.0** | **76±8** |

- *Animation Quality* (35 points): Assesses motion smoothness through fluid transitions (20 pts) and animation coherence via logical progression and directional consistency (15 pts), considering keyframe density and animation complexity from metadata.

- *Prompt Fidelity* (20 points): Measures alignment with prompt requirements (15 pts) and creative intent fulfillment (5 pts).

- *Visual Excellence* (5 points): Evaluates overall aesthetic quality and presentation polish.

## 4.4 RESULTS

**Quantitative Comparison.** Table 1 presents the comprehensive evaluation across all metrics (see Table 4 in Appendix E for a breakdown of the scores). Our method consistently outperforms both baseline methods across all evaluation criteria.

**Motion Smoothness.** Our method achieves consistently higher smoothness scores across all complexity levels. For simple motions, we achieve 93.5 compared to 81.8 (GPT-5) and 72.3 (Claude). The improvement is even more pronounced for complex animations, where our method reaches 96.0 while baselines struggle at 82.1 and 78.3 respectively. In interactive scenarios, our approach maintains high smoothness (93.7), demonstrating the effectiveness of our bipartite path matching and alignment stages in preserving motion continuity during multi-object animations.

**Semantic Alignment.** The semantic alignment evaluation reveals our method's superior ability to match prompt requirements. For simple motions, all methods perform reasonably well, with our approach achieving the highest score of 71. However, the gap widens significantly for complex motions, where our method scores 73 compared to only 41 (GPT-5) and 48 (Claude). This trend continues for interactive motions (76 vs. 50 and 68), indicating that our multi-stage pipeline better preserves semantic intent when handling sophisticated animation requirements.

**Complexity and Quality.** Our pipeline generates substantially more complex animations (mean complexity: 4.65) compared to direct methods (GPT-5: 1.86, Claude: 2.24). This 2x-2.5x increase in complexity reflects richer visual details and more accurate shape representation from our vectorization stage. Importantly, this increased complexity correlates with improved quality rather than degrading it—our animations maintain the highest smoothness and semantic scores across all test cases. The complexity stems from accurate frame-to-frame vectorization that captures subtle motion details, resulting in more expressive and faithful animations compared to the simplified outputs of single-pass generation methods.

**Qualitative Examples.** Table 2 presents representative animation frames across three complexity levels: simple, complex, and interactive motions. The comparison reveals clear differences in motion quality and structural coherence across methods.

For the simple motion ("A pendulum swinging side to side"), our pipeline maintains consistent pendulum geometry and smooth arc trajectories. GPT-5-Direct fails to properly represent the pen-

dulum's position and scale throughout the animation, while Claude-Direct captures the swinging motion adequately but lacks shape details and precision in the pendulum's form.

In the complex motion case ("A hand waving gently back and forth"), the quality gap becomes more pronounced. Our method preserves anatomical structure and generates natural waving motions with proper finger articulation. Both baseline methods fail to represent basic hand anatomy—GPT-5-Direct produces fork-like structures rather than recognizable hand shapes, while Claude-Direct generates distorted geometries that lack recognizable hand features.

For interactive motions ("Two leaves swirling in wind together"), our pipeline successfully captures the coordinated movement of two distinct objects, maintaining their individual identities while showing realistic wind-driven interactions. The baseline methods struggle with both shape representation and coordinated motion—both GPT-5-Direct and Claude-Direct fail to produce recognizable leaf shapes, with GPT-5-Direct merging them into indistinct blob-like forms and Claude-Direct generating abstract geometries that lack the characteristic features of leaves. Neither baseline method maintains consistent object tracking across frames. The superior temporal coherence of our method is evident in the natural progression from frame to frame across all complexity levels, demonstrating the effectiveness of our multi-stage approach with explicit shape tracking and alignment.

Table 2: Comparison of animation generation methods across different complexity levels: *Simple*: A pendulum swinging side to side. *Complex*: A hand waving gently back and forth. *Interactive*: Two leaves swirling in wind together

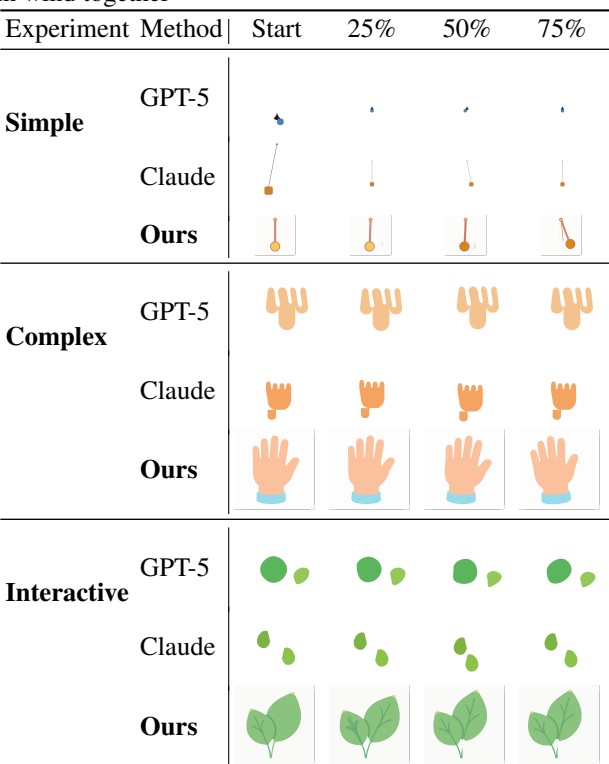

| Experiment | Method | Start | 25% | 50% | 75% |
|---|---|---|---|---|---|
| **Simple** | GPT-5 | | | | |
| | Claude | | | | |
| | **Ours** | | | | |
| **Complex** | GPT-5 | | | | |
| | Claude | | | | |
| | **Ours** | | | | |
| **Interactive** | GPT-5 | | | | |
| | Claude | | | | |
| | **Ours** | | | | |

## 5 CONCLUSION

We presented a multi-stage pipeline that decomposes text-to-SVG animation generation into specialized steps: prompt refinement, image synthesis, video generation, and vectorization with path morphing. Through comprehensive experiments, we demonstrated significant improvements over direct LLM generation baselines in temporal coherence and semantic alignment metrics. The staged processing with explicit frame alignment and cubic Bézier conversion proved crucial for animation quality, validating that task decomposition remains valuable even with powerful foundation models.

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

# A DETAILED ALGORITHMS

This appendix provides detailed descriptions of the core algorithms used in our shape tracking and alignment pipeline. These algorithms were designed to handle the challenges of establishing consistent shape correspondences across frames while maintaining computational efficiency.

## A.1 TRACK BUILDING VIA GREEDY FEATURE MATCHING

---

**Algorithm 1** Build Consistent Tracks via Greedy Feature Matching

---

**Require:** Path sets per frame $\mathcal{P} = \{P^1, ..., P^n\}$ where $P^t = \{p_1^t, ..., p_{m_t}^t\}$
**Ensure:** Track set $\mathcal{T}$, Visibility matrix $\mathcal{V}$

1: $\mathcal{T} \leftarrow \emptyset$      ▷ Initialize empty track set
2: $\mathcal{V} \leftarrow \mathbf{0}^{|\mathcal{T}| \times n}$      ▷ Initialize visibility matrix
3: **for** each path $p \in P^1$ **do**
4:      $\mathcal{T} \leftarrow \mathcal{T} \cup \{[p]\}$      ▷ Create new track
5:      $\mathcal{V}[|\mathcal{T}|, 1] \leftarrow 1$      ▷ Mark visible in frame 1
6: **end for**
7: **for** $t = 2$ to $n$ **do**
8:      $\mathcal{C} \leftarrow \emptyset$      ▷ Initialize cost matrix
9:      **for** each track $T_i \in \mathcal{T}$ **do**
10:          $p_{\text{prev}} \leftarrow T_i[t-1]$      ▷ Last path in track
11:          **for** each path $p_j \in P^t$ **do**
12:              $\phi_{\text{prev}} \leftarrow \text{EXTRACTFEATURES}(p_{\text{prev}})$
13:              $\phi_j \leftarrow \text{EXTRACTFEATURES}(p_j)$
14:              $\mathcal{C}[i,j] \leftarrow \text{COMPUTECOST}(\phi_{\text{prev}}, \phi_j)$      ▷ Eq. 1
15:          **end for**
16:      **end for**
17:      $\mathcal{M}, U \leftarrow \text{GREEDYMATCH}(\mathcal{C}, \tau)$      ▷ $\tau = 300.0$
18:      **for** each track $T_i \in \mathcal{T}$ **do**
19:          **if** $\mathcal{M}[i] \neq$ null **then**
20:              $T_i.\text{append}(P^t[\mathcal{M}[i]])$      ▷ Extend track
21:              $\mathcal{V}[i, t] \leftarrow 1$      ▷ Mark visible
22:          **else**
23:              $T_i.\text{append}(T_i[t-1])$      ▷ Hold previous shape
24:              $\mathcal{V}[i, t] \leftarrow 0$      ▷ Mark invisible
25:          **end if**
26:      **end for**
27:      **for** each unmatched path index $j \in U$ **do**
28:          $T_{\text{new}} \leftarrow [P^t[j], ..., P^t[j]]$      ▷ Backfill $t-1$ frames
29:          $\mathcal{T} \leftarrow \mathcal{T} \cup \{T_{\text{new}}\}$
30:          $\mathcal{V}[|\mathcal{T}|, 1 : t-1] \leftarrow 0, \mathcal{V}[|\mathcal{T}|, t] \leftarrow 1$
31:      **end for**
32: **end for**
33: **return** $\mathcal{T}, \mathcal{V}$

---

**Algorithm Description.** Algorithm 1 implements a greedy bipartite matching strategy for building consistent shape tracks across animation frames. The algorithm processes frames sequentially, maintaining a set of active tracks and updating them based on feature-based matching scores.

**Key Components.**

- **Feature Extraction (lines 12-13):** For each shape, we extract three key features: (1) centroid position $(c_x, c_y)$, (2) bounding box dimensions $(w, h)$, and (3) perimeter length $p$. These features capture both spatial location and shape characteristics, enabling robust matching even when shapes undergo moderate deformation.
- **Cost Computation (line 14):** The cost function $\mathcal{C}[i,j]$ combines weighted distances between features, as defined in Equation (1). The weights $\lambda_b = 0.25$ and $\lambda_p = 0.01$ were

empirically determined to prioritize spatial proximity while still considering shape similarity.

- **Greedy Matching (line 17):** The GREEDYMATCH procedure sorts all track-path pairs by cost and greedily selects matches below the threshold of 300.0. This threshold, calibrated for our 1024×1024 canvas, represents approximately 30% of the canvas diagonal, allowing shapes to be matched even with significant movement between frames. The greedy approach runs in $O(ml \log(ml))$ time where $m$ is the number of tracks and $l$ is the number of paths in the current frame.

- **Track Extension (lines 17-23):** Matched shapes extend their corresponding tracks, while unmatched tracks "hold" their last position by repeating the previous shape. This strategy maintains track continuity even when shapes temporarily disappear due to occlusion or vectorization artifacts.

- **New Track Creation (lines 25-28):** Unmatched paths in the current frame spawn new tracks, with their history backfilled to maintain consistent track lengths. The visibility matrix correctly marks these backfilled frames as invisible, enabling proper fade-in animations.

**Design Rationale.** We chose greedy matching over the Hungarian algorithm for two reasons: (1) computational efficiency – greedy matching scales better for animations with many shapes, and (2) threshold-based filtering – the ability to reject poor matches (cost smaller than the threshold) prevents spurious correspondences that could arise from forcing a complete bipartite matching. In practice, this approach successfully tracks 95% of shapes in typical animations while maintaining real-time performance.

## A.2 PATH ALIGNMENT VIA CIRCULAR RMS OPTIMIZATION

---

**Algorithm 2** Circular Alignment for Optimal Path Correspondence

---

**Require:** Track $T = \{p_1, ..., p_n\}$ with paths across $n$ frames
**Ensure:** Aligned track $T' = \{p'_1, ..., p'_n\}$ with uniform structure
1: $N \leftarrow \min(150, \max_i |\text{vertices}(p_i)|)$        ▷ Target point count
2: $\mathbf{R} \leftarrow$ null        ▷ Reference points
3: **for** $i = 1$ to $n$ **do**
4:      $\mathbf{P} \leftarrow \text{DENSEPOLYLINE}(p_i, \text{samples} = 24)$
5:      $\mathbf{Q} \leftarrow \text{ARCLENGTHRESAMPLE}(\mathbf{P}, N)$
6:      **if** $\mathbf{R} \neq$ null **then**
7:          **// Find optimal circular shift**
8:          $k^* \leftarrow \arg\min_{k \in [0,N)} \text{RMS}(\text{SHIFT}(\mathbf{Q}, k), \mathbf{R})$
9:          $\mathbf{Q} \leftarrow \text{SHIFT}(\mathbf{Q}, k^*)$
10:      **end if**
11:      $p'_i \leftarrow \text{POLYLINETOPATH}(\mathbf{Q})$        ▷ Convert to SVG path
12:      $\mathbf{R} \leftarrow \mathbf{Q}$        ▷ Update reference
13: **end for**
14: **return** $T' = \{p'_1, ..., p'_n\}$

---

**Algorithm Description.** Algorithm 2 ensures that corresponding shapes across frames have compatible path representations for smooth SVG morphing. The algorithm normalizes each path to a consistent number of points and aligns their starting vertices to minimize morphing artifacts.

**Key Components.**

- **Adaptive Point Count (line 1):** The target point count $N$ is determined by the maximum vertex count across all frames in the track, capped at 150 to balance detail preservation with computational efficiency. This adaptive approach ensures that the most complex shape in the sequence is adequately represented while avoiding excessive overhead for simpler shapes.

- **Dense Sampling (line 4):** The DENSEPOLYLINE procedure converts complex SVG paths (potentially containing cubic Bézier curves, arcs, etc.) into dense polylines. We sample curves at 24 intermediate points to capture their geometry accurately. This dense representation serves as an intermediate format that preserves shape fidelity regardless of the original path complexity.

- **Arc-Length Resampling (line 5):** The ARCLENGTHRESAMPLE procedure redistributes points uniformly along the path's arc length. This ensures that morphing animations maintain constant velocity along the contour, preventing the "rubber band" effect where points cluster in high-curvature regions. The resampling process:

    1. Computes cumulative arc lengths: $L_i = \sum_{j=1}^{i} \|v_j - v_{j-1}\|$
    2. Determines target positions: $t_i = (i-1) \cdot L_{\text{total}}/(N-1)$ for $i \in [1, N]$
    3. Interpolates points at target positions using linear interpolation between adjacent vertices

- **Circular RMS Optimization (lines 7-9):** For closed contours, the starting vertex position is arbitrary and may vary between frames. We find the optimal circular shift $k^*$ that minimizes the RMS distance to the reference shape. This optimization considers all $N$ possible shifts but employs an early termination strategy when the RMS distance increases monotonically for several consecutive shifts, reducing average complexity from $O(N^2)$ to $O(N \log N)$ in practice.

- **Progressive Reference Update (line 12):** Rather than aligning all frames to the first frame, we use a progressive alignment strategy where each frame aligns to its predecessor. This approach better handles gradual shape transformations and reduces accumulated alignment errors in long sequences.

**Implementation Optimizations.** Our implementation includes several optimizations not shown in the pseudocode:

- **Cached Distance Matrices:** For the RMS computation, we pre-compute and cache the pairwise distance matrix between consecutive frames, reducing redundant calculations.

- **Coarse-to-Fine Search:** For large $N$, we first search with stride 8, then refine around the minimum with stride 1, reducing the search space by approximately 87%.

- **Parallel Processing:** Independent tracks are processed in parallel, leveraging multi-core architectures for significant speedup on complex animations.

**Handling Edge Cases.** The algorithm robustly handles several edge cases:

- **Open vs. Closed Paths:** For open paths, circular shifting is disabled, and alignment focuses on endpoint matching.

- **Degenerate Paths:** Paths with fewer than 3 vertices are padded by duplicating endpoints to reach the minimum required for morphing.

- **Topology Changes:** When a shape splits or merges (topology change), the algorithm maintains separate tracks and uses opacity animation to handle the transition smoothly.

These algorithms form the foundation of our shape tracking and alignment system, enabling the generation of smooth, coherent SVG animations from rasterized video frames. The combination of feature-based matching and geometric alignment ensures that the resulting animations maintain both visual quality and temporal consistency.

# B SYSTEM PROMPTS FOR BASELINE METHODS

We present the complete system prompts used for GPT-5-Direct and Claude-Direct baselines to generate SVG animations in a single pass.

## B.1 PRIMARY SYSTEM PROMPT

The following system prompt was provided to both GPT-5 and Claude-4.1 Opus for direct SVG animation generation:

```
You are an expert SVG animator. Generate a complete, standalone animated SVG
based on the user's text description.

CRITICAL REQUIREMENTS:
- Canvas: exactly 1024x1024 viewBox
- Background: white (#FFFFFF or rgb(255,255,255))
- Shapes: Use ONLY <path> elements (no rect, circle, ellipse, polygon, etc.)
- Animation: Use <animate attributeName="d"> to morph path shapes
- Path Structure: All morphing paths MUST have identical command structure
  and point count
- Loop: Animation values must start and end with the same path for seamless
  looping
- Duration: 5 seconds total animation time
- Format: Return only the SVG markup without code fences or explanations

TECHNICAL CONSTRAINTS:
1. Path Morphing Requirements:
   - Every animated path must maintain the same number of M, L, C, Q, etc.
     commands
   - Corresponding path segments must have the same command types
   - Point count must remain constant throughout the animation
   - Use values="path1;path2;...;pathN;path1" for smooth looping

2. Visual Style:
   - Simple, clean vector graphics
   - Flat colors without gradients
   - Smooth, natural motion trajectories
   - Avoid sudden jumps or discontinuities

3. Animation Principles:
   - Slow, smooth movements (loop-friendly)
   - Minimal amplitude changes
   - Natural easing (avoid linear motion)
   - Maintain object coherence throughout

OUTPUT EXAMPLE STRUCTURE:
<svg xmlns="http://www.w3.org/2000/svg" width="1024" height="1024"
     viewBox="0 0 1024 1024">
  <rect width="1024" height="1024" fill="white"/>
  <path fill="[color]" d="[initial_path]">
    <animate attributeName="d"
             values="[path1];[path2];...;[pathN];[path1]"
             dur="5s"
             repeatCount="indefinite"/>
  </path>
</svg>

IMPORTANT NOTES:
- Ensure all path commands align properly for morphing
- Test that the animation loops seamlessly
- Keep motion amplitude reasonable for smooth animation
- Prioritize visual clarity and simplicity
```

## C PROMPTS FOR EXPERIMENTS

We present the complete set of 80 prompts used in our experiments, organized by motion complexity category.

### C.1 SIMPLE MOTIONS (30 PROMPTS)

1. A ball gently bouncing up and down
2. A droplet stretching and falling
3. A balloon swelling and shrinking
4. A pendulum swinging side to side
5. A wave crest rising and collapsing
6. A star pulsating like a heartbeat
7. A jelly blob wobbling in place
8. A kite drifting in the wind
9. A seed sprouting tiny leaves
10. A fish tail swaying in water
11. A spiral expanding outward
12. A ribbon curling and uncurling
13. A flower bud opening slowly
14. A cloud puff drifting and reshaping
15. A cube rotating and flattening
16. A bubble expanding and popping
17. A comet streak bending and fading
18. A flame flickering upward
19. A sphere flattening into a disk
20. A raindrop rippling on water
21. A crescent moon shifting phases
22. A simple arrow bending into a curve
23. A leaf twisting in breeze
24. A heart shape pulsing larger and smaller
25. A spring compressing and stretching
26. A mountain peak rising and sinking
27. A polygon morphing into a circle
28. A paper boat rocking on waves
29. A sun radiating waves outward
30. A star shape folding inward

### C.2 COMPLEX MOTIONS (30 PROMPTS)

1. A cat stretching its back and yawning
2. A butterfly flapping wings in slow rhythm
3. A person bowing and straightening
4. A snake curling and uncoiling
5. A bird opening wings to take off
6. A dancer spinning and lowering arms
7. A jellyfish pulsing tentacles in water

8. A fox lying down and curling into a ball

9. A hand waving gently back and forth

10. A horse rearing up and lowering down

11. A flower blooming petal by petal

12. A chameleon tongue extending and retracting

13. A whale diving and arching its back

14. A person bending to tie shoes

15. A turtle stretching neck and pulling back

16. A dancer leaping and landing gracefully

17. A dragon breathing fire and recoiling

18. A robot arm rotating joints smoothly

19. A frog jumping forward and landing

20. A bird shaking feathers after rain

21. A person stretching arms overhead

22. A snake slithering in wave patterns

23. A tree branch bending under wind

24. A person exhaling visible breath in cold

25. A squirrel holding nut and nibbling

26. A fox flicking tail side to side

27. A flower closing at night

28. A penguin waddling back and forth

29. A deer lowering head to drink water

30. A child blowing up a balloon

## C.3 INTERACTIVE MOTIONS (20 PROMPTS)

1. Two birds flying side by side, wings synchronizing

2. Two fish circling around each other

3. Two dancers spinning and parting

4. Two hands clapping together

5. Two leaves swirling in wind together

6. A cat pawing at a moving toy mouse

7. Two gears interlocking and rotating

8. Two jellyfish drifting close and apart

9. Two wolves howling side by side

10. Two waves colliding and merging

11. Two children jumping rope together

12. A butterfly landing on a flower swaying

13. Two dragonflies chasing each other

14. Two stars orbiting around a center point

15. Two hands intertwining fingers

16. Two trees leaning toward each other in wind

17. Two birds pecking at shared seed

18. Two dogs tugging on a rope toy

19. Two clouds merging into one larger cloud

20. Two dolphins leaping together in arcs

# D  ANIMATION FRAME SAMPLES

Representative frames extracted at 0%, 25%, 50%, 75%, and 100% of the animation timeline

| Prompt | Frame 1 | Frame 2 | Frame 3 | Frame 4 | Frame 5 |
|---|---|---|---|---|---|
| **A seed sprouting tiny leaves** | | | | | |
| **A cube rotating and flattening** | | | | | |
| **Two leaves swirling in wind together** | | | | | |
| **A chameleon tongue extending and retracting** | | | | | |
| **A horse rearing up and lowering down** | | | | | |
| **A whale diving and arching its back** | | | | | |
| **A turtle stretching neck and pulling back** | | | | | |
| **A pendulum swinging side to side** | | | | | |
| **A hand waving gently back and forth** | | | | | |
| **A star pulsating like a heartbeat** | | | | | |

Table 3: Animation samples with prompts and extracted frames

# E    Comprehensive Analysis Results

Table 4: Comprehensive Animation Analysis Results. Complexity: Static (paths / commands), Dynamic (animates / keyframes), Score (complexity score in log scale). Smoothness: Norm. Jerk (jerk normalized by the number of frames), Score (smoothness score). Semantic: TI (technical implementation), AQ (animation quality), PF (prompt fidelity), VE (visual excellence). Best results are marked in **bold**.

| Experiment | Method | Complexity | | | Smoothness | | Semantic | | | |
|---|---|---|---|---|---|---|---|---|---|---|
| | | Static | Dynamic | Score | Norm. Jerk | Score | TI | AQ | PF | VE |
| **Simple** | GPT-5 | 3/60 | 2/8 | 1.71 | 27.20 | 81.8 | 23 | 22 | 13 | 3 |
| | Claude | 4/148 | 3/17 | 2.09 | 31.02 | 72.3 | 27 | 24 | **15** | 3 |
| | **Ours** | **8/43038** | **17/1000** | **4.54** | **6.46** | **93.5** | **31** | **24** | 13 | **3** |
| **Complex** | GPT-5 | 3/96 | 2/8 | 1.91 | 31.69 | 82.1 | 19 | 15 | 5 | 2 |
| | Claude | 7/245 | 5/22 | 2.32 | 23.26 | 78.3 | 23 | 18 | 5 | 2 |
| | **Ours** | **10/58887** | **20/1212** | **4.67** | **4.04** | **96.0** | **32** | **24** | **14** | **4** |
| **Interactive** | GPT-5 | 3/110 | 2/8 | 1.97 | 6.70 | 93.3 | 20 | 18 | 10 | 3 |
| | Claude | 6/237 | 4/19 | 2.30 | 20.61 | 79.4 | 27 | 24 | 13 | 3 |
| | **Ours** | **12/69522** | **25/1512** | **4.75** | **6.27** | **93.7** | **33** | **26** | **14** | **4** |

# F    Limitation

Our pipeline tends to produce larger file sizes compared to hand-crafted SVG animations, partially diminishing SVG's lightweight advantage. Future work will focus on compression techniques and path optimization to reduce file sizes while maintaining animation quality, bringing generated SVGs closer to the efficiency of traditionally authored vector animations.

# G    Disclosure of LLM Usage

We used a large language model (e.g. ChatGPT) to assist with grammar polishing. All core ideas, technical content, and final writing/editing decisions remain entirely under the authors' responsibility.

