# OpenReview forum: "SOLA: Text-based animated vector graphics generation with agentic orchestration"
_ICLR.cc/2026/Conference — Submitted to ICLR 2026_

### Official Review · Reviewer_3aVh · 2025-10-23

**Soundness:** 2
**Presentation:** 3
**Contribution:** 1
**Rating:** 2
**Confidence:** 4

**Summary:**

The paper proposes SOLA (SVG-Oriented Language-to-Animation), an agentic pipeline for generating animated SVGs directly from natural language prompts.  It orchestrates a multi-stage workflow using pre-trained models (text-to-video, image generation, vectorization) coordinated via a LangGraph agent to produce resolution-independent SVG animations. The pipeline includes stages for prompt augmentation, video synthesis, vectorization, greedy bipartite shape tracking, path normalization, and smooth morphing. A new evaluation protocol with an 80-prompt test set and metrics for complexity, smoothness, and semantic alignment is introduced. Experimental results demonstrate that the proposed workflow outperforms leading LLM-based baselines in generating semantically faithful, temporally smooth, and complex vector animations.

**Strengths:**

1. SOLA presents a convincingly modular and training-free approach to text-to-SVG animation generation, leveraging existing pre-trained models and orchestrating them in an agentic pipeline without requiring new large-scale annotated data or additional model training. This is practical given the lack of SVG animation datasets.
2. The use of a feature-based greedy bipartite matching for shape correspondence across frames (Section 3.5, Equation for $C_{ij}$), followed by path normalization with arc-length parameterization and RMS-based circular alignment (Algorithms 1 and 2), shows careful algorithmic handling of vector morphing, which is essential for temporal coherence.
3. Results in Table 1 and Table 4 consistently show that SOLA improves both animation smoothness and complexity over leading LLM-based direct methods across all tested motion categories, often by significant margins, which is also evident in frame-by-frame qualitative samples (Figure in Table 3, img-1.jpeg).

**Weaknesses:**

1. I personally do not recognize this paper as a research work. The key process is realized by VTracer, an open-source software, to convert the generated rasterized image into SVG. The animation procedure is also trivial, using Bipartite Matching to track the paths. I cannot find novelty in this paper.
2. The baselines are single-pass LLMs that directly generate SVG codes, which is a definitely different way to generate SVG. The comparison to the baselines is unfair.
There is no clear ablation study showing the quantitative impact of individual components (e.g., what if bipartite matching is replaced by a simpler assignment, or if Bézier smoothing is omitted).

**Questions:**

Is the system robust to prompts substantially out-of-distribution from your 80-sample test set (e.g., highly abstract, surrealist, or text-centric prompts)? Can you share examples and discuss generalizability?

---

### Official Review · Reviewer_pUzu · 2025-10-26

**Soundness:** 3
**Presentation:** 3
**Contribution:** 2
**Rating:** 4
**Confidence:** 3

**Summary:**

This submission describes an automatic workflow to create animated SVG. They first use text-to-image models to create a static frame given user prompt, then use image-to-video generation to create SVG-styled video. Then they use VTracer to vectorize the frames, and further optimize the SVG with shape alignment, path tracking, path morphing, and bezier curve fitting. The whole pipeline is orchestrated with LangGraph. Finally, to evaluate the proposed workflow, they created an animated SVG generation benchmark, with two baseline methods (GPT-5, Claude), 80 prompts, and 3 metrics (complexity, smoothness, semantic score). The proposed method outperforms two baseline methods, which directly generate the final SVG file from the given prompt, in all three metrics.

**Strengths:**

+ The proposed workflow is intuitive. The text-to-image and image-to-video generation model clearly have better ability in visual content generation. It makes sense to utilize these models to assist the SVG creation.
+ The process of converting video frames to animated SVG consists of many useful algorithms, which are also intuitive and might be handy tools for the community of SVG creation.
+ The proposed SVG creation benchmark is a nice contribution, as a supplement for the LLM ability evaluation. Clearly, current LLMs still fall short in animated SVG generation.

**Weaknesses:**

- The novelty and technical contribution of the proposed pipeline is weak. SVG generation is not a new task for LLMs. The whole workflow is a simple combination of multiple models. I still think this workflow is valuable for the content creation community. It is just I am not sure whether this kind of work aligns with the standard of ICLR.
- Some other possibilities of baseline. Since the proposed baseline makes use of text-to-image/video generation ability, maybe it also makes sense to feed those generated image/video frames to multi-modal LLMs and prompt them to generate animated SVG files corresponds to the input image/video frames. Or we could ask GPT to first generate images, then in the same context, ask it to generate SVG files.

**Questions:**

N/A

---

### Official Review · Reviewer_XRKQ · 2025-11-01

**Soundness:** 3
**Presentation:** 2
**Contribution:** 3
**Rating:** 4
**Confidence:** 3

**Summary:**

The paper introduce SOLA, which is a zero-shot, agentic pipeline that turns text into animated SVGs by chaining pretrained models, then enforcing shape tracking, path normalization, and Bezier fitting. On 80 prompts, it beats GPT-5/Claude direct SVG baselines in complexity, smoothness, and semantic alignment.

**Strengths:**

1. It is the first zero-shot, agentic workflow to produce animated SVGs directly from text.

2. The proposed method is robust to handles occlusion paths via track visibility and repetitions; or topology changes managed with separate tracks/opacity transitions.

3. Under an 80-prompt protocol, reports better complexity, smoothness, and semantic alignment than direct LLM-to-SVG approaches.

**Weaknesses:**

1.. The 80-prompt study is relatively small and omits comparisons to recent learned vector-animation models; user studies and task-specific metrics are suggested.

2. The method has not proven the stability across different backbones. How is the robustness and effectiveness when changed to another backbone? How does the backbone capability influence the final output.

3. Why there is no conventional baseline comparisons to provide an assessment of the quality of former traditional pipeline? Does the animation generated by this agentic workflow really better than former ones. A detailed justification is needed.

4. The paper proposes greedy matching, which favors speed. It may degrade with heavy crowding/fast motions.

**Questions:**

1. Look for a larger-scale tests, comparisons to recent learned vector-animators, task-specific metrics, and a reasonable user study.

2. Report results on latest and alternative T2V/raster backbones; analyze how backbone quality affects SVG fidelity and failure modes.

3. Include strong traditional pipelines and ablations to justify superiority of the agentic workflow.

4. Provide stress tests with runtime–quality trade-offs.

---

### Official Review · Reviewer_JJdv · 2025-11-01

**Soundness:** 2
**Presentation:** 2
**Contribution:** 2
**Rating:** 2
**Confidence:** 4

**Summary:**

This paper presents SOLA (SVG-Oriented Language-to-Animation), a pipeline for generating animated SVG graphics from natural language descriptions. The system employs an agentic LangGraph workflow: (1) prompt refinement for image and video generation, (2) video synthesis using Gen-4, (3) frame vectorization via VTracer, (4) shape correspondence through greedy bipartite matching based on geometric features, (5) path normalization using arc-length resampling and circular alignment, and (6) cubic Bézier curve fitting. To evaluate the approach, the authors construct an 80-prompt test set spanning three complexity levels and propose metrics for animation complexity, motion smoothness (jerk-based), and semantic alignment (LLM-as-Judge). Experiments compare SOLA against GPT-5-Direct and Claude-Direct baselines, showing improvements across all metrics.

**Strengths:**

The paper pioneers a new research direction by tackling end-to-end text-to-animated-SVG generation, filling a clear gap between static vector generation and animation synthesis. The evaluation framework is comprehensive, providing 80 prompts across three complexity levels with multi-dimensional metrics that will benefit future research.

**Weaknesses:**

1. The baseline comparison is unfair—SOLA leverages Gen-4 trained on massive video datasets while baselines generate from text, essentially comparing "video generation + format conversion" versus "code generation capability."

2. The technical innovation is limited as core algorithms are standard techniques, and the hyperparameters used lacking justification. The "agentic orchestration" terminology overstates what is essentially a sequential pipeline where LangGraph appears unnecessary. Critical analyses are missing including failure cases, computational costs, and comparison with traditional vector animation workflows.

**Questions:**

1. Could you provide fair baselines that also leverage generative models, such as applying Keyframer to GPT-Image-1 generated static SVGs, or extending VectorFusion with temporal modeling? This would enable proper method-to-method comparison rather than comparing against pure code generation.

2. Can you provide an ablation study isolating Gen-4's contribution? For example, what happens if you replace Gen-4 with simpler video generation models—how much performance comes from the video model versus your shape tracking pipeline?

3. Can you provide human evaluation results to validate the automated metrics? Given the high variance (±8-12 points) and potential circular bias from using GPT-5 as both baseline and judge, human assessment is critical for establishing credibility.

---

### Official Review · Reviewer_8AFo · 2025-11-01

**Soundness:** 3
**Presentation:** 3
**Contribution:** 3
**Rating:** 4
**Confidence:** 4

**Summary:**

SOLA introduces a zero-shot, agentic pipeline for text-to-animated SVGs: prompt refinement, text-to-video, frame vectorization, greedy shape matching, circular RMS alignment, SMIL morphing, and Bézier fitting.

**Strengths:**

1. This paper aims at a new task, text-to-SVG generation, which is novel and intriguing.

2. The authors proposed a comprehensive benchmark for the new proposed task, which contributes a lot to the future research.

3. The paper is well-written, ensuring that its content is easily understandable for readers.

**Weaknesses:**

1. The paper targets animation generation but provides no supplementary demo video to showcase prompts and outputs. Beyond objective metrics, subjective user-preference results should also be reported. A comprehensive benchmark is needed.

2. Necessary ablations are missing. The paper only compares with base GPT-5/Claude, without ablating its own components, so each module’s effectiveness is unsubstantiated.

3. It would be better to report inference speed and token usage.

4. For each component, it should be better to include finer-grained experiments exploring hyperparameter settings.

**Questions:**

Please kindly refer to the weaknesses mentioned above.

---

### Meta-Review · Area_Chair_Af88 · 2025-12-26

**Summary:**

All the reviewers gave negative scores to this paper, while no rebuttal was submitted. So, it indicates a consisus to reject this paper.

**Reviewer Concerns:**

summary of concerns that still exist:

Limited Technical Novelty: Reviewers argue the work is a straightforward combination of existing tools (e.g., VTracer, Bipartite Matching) rather than a novel research contribution, questioning if the "agentic" pipeline is just a simple sequential workflow.

Unfair Baseline Comparisons: The evaluation is criticized for comparing a complex video-to-SVG pipeline against single-pass LLMs that generate code directly from text, essentially comparing different fundamental tasks.

Insufficient Evaluation & Ablation: The paper lacks internal ablation studies to justify specific modules, fails to test robustness across different model backbones, and uses a relatively small prompt benchmark (80 prompts).

Missing Comparative Analysis: There is a notable absence of comparisons against traditional vector animation workflows, recent learned SVG models, and human user-preference studies.

Lack of Empirical Rigor: Reviewers pointed out the absence of demo videos (essential for animation tasks), missing data on inference speed and token costs, and a lack of failure case analysis.

**Reviewer Scores:**

All reviewers would remain their initial scores

---

### Decision · Program_Chairs · 2026-01-26

Reject